# Longitudinal Patterns of Grandchild Care in South Korea

**DOI:** 10.3390/ijerph19031136

**Published:** 2022-01-20

**Authors:** Aely Park

**Affiliations:** Department of Social Welfare, Sunchon National University, Suncheon 57922, Korea; apark@scnu.ac.kr

**Keywords:** longitudinal patterns of grandchild care, grandparenting, Korean Longitudinal Study of Aging, latent trajectories

## Abstract

This study examined the longitudinal patterns of grandchild care to observe the influence of factors related to social participation, financial support to grandparents, demographic characteristics, and family structure on classifying the grandchild care. The rate of grandparent care for grandchildren was increasing, and the amount of time commitment for grandchild care was large in South Korea. Understanding how grandchild care unfolds over time and who is likely to provide ongoing grandchild care helps to advance the knowledge about grandparents providing grandchild care. The total sample consisted of 333 South Korean grandparents derived from the 3 waves of nationally representative data. This study utilized growth mixture modeling to identify latent classes of longitudinal patterns of grandchild care, and ran a multinomial logistic regression to examine the relationships between factors related to grandparents, adult children, and family structure and the identified latent classes. Grandchild care was classified into one of three categories: low-level decrease, high-level decrease, and low-to-high increase. Grandparents in the group of low-to-high increase were more likely to have higher financial dependence on adult children and have lower social participation than grandparents in other groups. Findings indicate that there are distinct subgroups among grandparents who care for their grandchildren. Additionally, those in the three classifications varied according to financial support received from adult children, social participation, and personal and family structure. Our findings inform policymakers to provide older adults a means to maintain their self-sufficiency. The community needs to provide programs and resources for working parents on behalf of grandchild care.

## 1. Introduction

Grandparenting has been receiving considerable attention in South Korea in recent years [1,2]. Grandchild care has become increasingly common due to a shortage of options for working parents who seek quality childcare in South Korea [3]. Indeed, employed mothers want to send their children to qualified childcare facilities, such as licensed providers [4], but such facilities only have capacity for less than 5% of the children who need care in South Korea [5]. Consequently, working mothers with infants often utilize informal childcare, which is predominantly provided by grandparents in South Korea [3]. In this study, childcare provided by grandparents refers to grandparents who do not have custody of their grandchildren but are caregivers for their grandchildren while the parents go to their places of employment, then return the grandchildren to their parents’ care following work each day.

Approximately 30% of South Korean grandparents who have adult children going to their places of employment have provided grandchild care at some point in their lives [2], and South Korean grandparents who provide care for their grandchildren report doing so for an average of 52 h per week [1]. This form of extended intergenerational care is not necessarily new; traditionally, grandchildren in South Korea were raised by their grandparents and parents in extended family systems, and adult children took responsibility to care for their older parents out of filial piety [6,7]. However, these traditional extended family interdependencies have been largely supplanted by nuclear families over the past century. In the early 1900s, approximately 60% of South Korean families lived in extended families, but by 1960, approximately 65% of South Korean families lived in nuclear family systems, and this rate further increased to 76% in 1990 and 82% in 2010 [8].

Before the 1990s, most mothers in nuclear families were not employed for pay outside of their homes and took care of their children at home. However, women’s participation in the job market rapidly increased following the economic depression of 1997, and working mothers began to use informal childcare provided by grandparents [9]. At that time, public daycare was typically used by low-income families, and particularly single-parent families who also depended on government assistance. Private childcare centers existed, and services were offered to the general families, but the services did not meet the needs of working mothers [9].

Strikingly, but perhaps not surprisingly given its rarity prior to the late-1990s, no research was published on grandchild care in South Korea throughout most of the 1990s. The first study on this topic was published in 1999 using a small community sample. It found that the emergence of grandchild care was related to women’s participation in the job market [9]. South Korean women, including mothers, have continued participating in the workforce at ever-increasing rates over the past few decades [10], and the involvement of grandparents in caregiving is on the rise in this modern scenario of maternal employment and inadequately qualified childcare facilities for their children [1,2].

To ensure that working women are compatible with work and childcare, South Korea’s policymakers have begun to pay attention to childcare, and particularly formal childcare to support working mothers. In 2012, the government introduced new public policies such as a childcare voucher program that allows preschool children to use public or private daycare centers for free [11]. In addition, parents who do not or cannot use the free childcare service receive a small allowance to be used toward childcare costs. After the voucher program took effect, free childcare service users (mostly non-working mothers) increased by nearly 12%, and private daycare use increased by approximately 8% [12], but private centers, which comprise the majority of daycare facilities, also prefer to take in children of non-working parents because those parents tend to pick up their children before 5 pm [4]. In short, the services of daycare facilities are typically inadequate for parents who work non-traditional hours (i.e., evenings, nights, weekends), and sometimes their availability is even inadequate for parents who work daytime hours but are unable to pick their children up before 5 pm.

Given these challenges associated with daycare facilities, many parents have turned to informal childcare from grandparents, and they often pay the grandparents, even if much less than the private care would cost at a daycare center [1,13,14]. Although grandparents have traditionally been taking care of extended families [15], caring for grandchildren often inhibits grandparents’ ability to prepare for later life these days [16]. For example, one study found that South Korea grandmothers who had continuously raised a grandchild were less economically, socially, and physically prepared for retirement than those who did not raise a grandchild [17]. In addition, high involvement in grandchild care can cause physical or emotional hardships and constrain social participation. For example, time spent providing grandchild care is positively correlated with one’s level of depressive symptoms, particularly for grandmothers [16]. That is, grandchild care may conflict with individuals’ expectations of active ageing, such as continued participation in social and economic affairs as a means to maintain autonomy and independence [17,18]. Although policymakers in South Korea have started to consider the establishment of benefits for those grandparents, there are currently no assistance programs for grandparents who take care of their grandchildren.

Despite evidence that grandchild care is associated with some undesirable outcomes for grandparents, few studies have investigated longitudinal patterns in grandchild care that may predict those outcomes. For example, there are at least two studies identifying patterns of grandchild care in South Korean and Taiwan samples—continuing, stopping, or restarting care. They found that grandparents who stopped caring for their grandchildren had a higher risk of developing depressive symptoms than grandparents who did not care for their grandchildren [19,20]. They explained that the harmful effect may be related to feelings of loss or detached from norms of intergenerational bonds and relationships. However, very few empirical studies have investigated grandchild care from a long-term perspective [2,21]. An inherent problem in prior literature on grandparenting is that studies are typically based on one-time interviews with older adults, and studies that take this static approach cannot examine longitudinal patterns in grandparent caregiving. In an attempt to fill this gap in the literature, this longitudinal study was designed to identify and advance understanding of subgroup differences among grandparents who care for their grandchildren, and to identify factors that predict membership in the distinct subgroups of grandchild care. Understanding how grandchild care unfolds in South Korea over time and who is likely to provide ongoing grandchild care can help to advance knowledge about grandparent caregivers and provide direction for designing assistance programs to meet their needs.

### 1.1. Factors Predicting Grandchild Care

#### 1.1.1. Grandparent Factors

Among grandparents, evidence suggests that grandparent age, gender, education, and health are related to patterns of care provided for grandchildren. Younger grandparents are more likely to provide caregiving for their grandchildren than older grandparents, and among those who do provide such care, younger grandparents tend to spend more time caring for their grandchildren than do older grandparents [1,22,23]. With regard to gender, Lee and Bauer [1] found that grandmothers were more likely than grandfathers to participate in grandchild care, and this was the case across cultures. In South Korea, grandmothers’ involvement in grandchild care is approximately three times greater than that of grandfathers [2]. In particular, grandmothers who provide extensive care tend to do so for roughly 50 h per week [1,3].

Findings concerning the association between grandparents’ education and caregiving patterns for grandchildren are more nuanced. Some studies have reported that older adults who provide grandchild care are more likely to be well-educated than those who do not [23,24], but others have reported that older adults who provide extensive care are less likely to be well-educated than those who either do not provide care or who only do so occasionally [21,22,25]. Further, Lee and Bauer [1] did not find a relationship between grandparents’ education and the number of hours they spent caring for their grandchildren. A plausible explanation for these mixed results is that well-educated older adults have a strong desire to help their children succeed socially and economically, and thus they are willing to care for their grandchildren, but well-educated older adults are more likely to provide occasional rather than extensive care [26]. Regarding health, grandparents who provide any care for their grandchildren tend to be healthier than those who do not [23,24,27], but there does not seem to be a relationship between grandparents’ health and the number of hours spent caring for grandchildren [1]. Importantly, however, less is known about longitudinal patterns of grandchild care—that is, how grandparents’ caregiving for grandchildren changes over time within the same family—because most studies conducted in this area to date have used cross-sectional data.

Additionally, social participation seems to play a role in predicting the quality of life for older adults—older adults tend to spend most of their time in social gatherings, social networking, and various social activities, which enhances quality of life [28]. Research to date has been mixed with regard to the direction of the association between social participation and providing care for grandchildren. Social participation has been found to be associated with a higher likelihood of providing grandchild care and with providing more grandchild care [1,22,24,27]. In contrast, however, some other studies have found that older adults who provide grandchild care have difficulty maintaining social activities due to a lack of time [1,29,30,31].

#### 1.1.2. Adult Children Factors

The intergenerational transfer of resources between parents and their adult children can be discussed within the boundaries of social exchange theory [1,7,13,15]. Asian grandparents provide grandchild care for their adult children as part of a strong norm of mutual exchange dynamics in families [32]. Even in a different sample, Asian grandparents tend to perceive grandparenting as being related to fostering intergenerational bonds and supporting their adult children’s career success [7]. Adult children also feel safe leaving their children in their parents’ care [33] and value the opportunity for intergenerational bonding to occur [7]. They receive financial support from their adult children for the children’s care, and further, grandparents who provide extensive care are more likely to expect financial support from their adult children than grandparents who provide occasional care [1,13]. Thus, grandparents might spend different amounts of time doing grandchild care based on their expectations of financial (or caregiving) support from their adult children. Grandparents sometimes give up social participation in favor of caring for their grandchildren, and do so in exchange for mutual bonding, financial support, and being cared for when they need it. Reciprocally, from a social exchange perspective, adult children are able to obtain trusted and flexible childcare at a low cost, relative to formal care, provide some immediate financial assistance to their parents, and facilitate the building of intergenerational bonds between their parents and their children.

Adult children whose parents provide childcare tend to expect that childcare will cost them less than if their parents did not provide childcare [13,34]. Whether conceptualized as deferred compensation or reciprocity, a social exchange framework explains this context well. Specifically, those discounted childcare services appear to be reimbursed later because adult children are more likely to provide financial support for grandparents who cared for their grandchildren than for those who did not perform any caregiving tasks [35,36].

With regard to characteristics of the adult children (parents of the grandchildren), one study found that the parents of younger mothers were more likely to provide intensive grandchild care than were the parents of older mothers [37]. However, it is unclear whether this was due to characteristics of the younger mothers, an association between mother age and grandmother age, or characteristics of younger families (e.g., fewer years removed from the original parent–child relationship). 

#### 1.1.3. Family Structure Factors

Family structure factors that have been identified in the literature as predictors of grandparent caregiving for grandchildren include co-residence, number of generations in households, and number of people in households. Evidence indicates that caring for grandchildren is more common [24] and consumes more hours [37] when grandparents and their adult children live in close proximity to one another. In South Korea, compared to grandparents who do not care for their grandchildren, those who do provide such care are more likely to live in three-generation households comprised of grandparents, adult children, and grandchildren [23]. This pattern is also found in the United States, where a growing number of children have lived with a grandparent—the proportion increased from 3% in 1970 to 7% in 2010 [38].

### 1.2. Research Questions

The present study was designed to examine the longitudinal patterns of grandchild care over a long-term period. This was accomplished using the general growth mixture modeling approach, which is an advanced way to identify distinct subgroups of grandparents who follow similar trajectories of grandparenting over time. Based on empirical and theoretical literature, the following research questions (RQ) guided this study:

RQ1. Are there distinct longitudinal patterns of grandchild care across a five-year period?

RQ2: Are social participation, financial support to grandparents, demographic characteristics, and family structure associated with the longitudinal patterns of grandchild care?

## 2. Method

### 2.1. Data and Sample

For this study, data were used from the Korean Longitudinal Study of Aging (KLoSA), which contains information on income and asset status, retirement decisions, grandparenting, health, and intra-family transfer of income across four waves of data collected biannually. Wave 1 (2006) was comprised of a nationally representative sample of 10,000 individuals from 6171 households, who were recruited using two-stage area probability sampling. Data were collected from all persons 45 years of age or over who resided in sampled households. Face-to-face interviews were also conducted by trained interviewers, who visited the selected households and collected data using computer-assisted personal interviewing techniques.

For the present study, we used Wave 2 (2008), Wave 3 (2010), and Wave 4 (2012) of the individual-level data because the KLoSA began to include key variables for our study in Wave 2, such as amount of time spent caring for each grandchild. On that point, our sample is limited to grandparents who cared for grandchildren younger than ten years of age because the KLoSA defined grandchild care as caring for grandchildren younger than age ten. Additionally, grandparents who were aged between 50 and 70 at the time of the interviews in Wave 2 were selected to include younger grandchildren who needed to be cared for. Predictors for grandparenting time in Wave 2 and dependent variables in Waves 2–4 were used to establish a temporal order. As a first step, out of the 10,000 individuals who participated in the study, a total of 333 grandparents who reported providing grandchild care for at least one hour per week across each of the three waves were selected for analyses. When a grandmother and grandfather who resided in the same household both provided care (*n* = 70 couples), both were included in the analyses. Since KLoSA did not provide information on the primary caregiver of grandchildren when both grandmother and grandfather provided grandchild care, we did not arbitrarily exclude the data reported by grandmothers or grandfathers from the analysis. For parsimony, given that many respondent characteristics are also key analytical variables, sample characteristics are described at the start of the Results Section.

In terms of respondent attrition, the KLoSA had lost roughly 26% of the focal subjects by Wave 4, and this study examined whether the attrition in Wave 4 was different based on sample characteristics including grandparenting indicators; to check the attrition, this study used sample participants who were aged between 50 and 70 at the time of the interviews in Wave 2 (*n* = 5947). This study measured the prevalence of demographic and grandparenting indicators for the full sample (*n* = 5947) from Wave 2, those who remained (*n* = 5228) in Wave 4, and those who did not remain (*n* = 719) for Wave 4. Based on Pearson chi-square and t-tests, there were no statistically significant differences between the groups that did versus did not return for Wave 4 based on their demographic characteristics, such as gender, age, high school graduation, and household income, as well as grandparenting factors such as whether or not they cared for their grandchildren and how much time they spent doing so. 

### 2.2. Measures

#### 2.2.1. Outcome

The KLoSA did not collect information about the exact time schedules for grandchild care, only the amount of time spent with each grandchild was collected, which was measured with one item: “How many hours per week did you spend caring for a grandchild in an average week during the past one year?” This was measured on a per-grandchild basis, and 46% of respondents reported caring for more than one grandchild. Therefore, to avoid double-counting overlapping hours spent with multiple grandchildren, this study only used the largest amount of time reportedly spent with any one grandchild. Additionally, because this was self-report data, caution should be exercised when interpreting these data given that there could be variation in how time for care was calculated and reported across respondents. For example, the calculation may have been more straightforward for those not living with their adult children (and presumably grandchildren) than for those who were living with them.

#### 2.2.2. Independent Variables

Several predictors identified in Wave 2 were included in this study. For example, the characteristics of the primary respondents who cared for their grandchildren such as their age, gender, educational attainment, health status, and social participation were included. Additionally, this study included characteristics of adult children such as their age and financial support, and family structure factors such as co-residence, number of generations, and number of people in the household.

Four personal characteristics of the grandparents plus one social exchange variable were included. This study measured age as a continuous variable using the year of the participant’s birth. Gender was a dichotomous variable, limited to male (coded as 1) and female (0). Education was also categorized dichotomously, as either not graduated from high school (0) or graduated from high school (1). Health status was measured with one self-report question: “What is your general health status?” Response options ranged from very healthy (1) to very unhealthy (5). Finally, social participation was measured by asking whether the respondent (grandparent) had participated in any among a list of six broadly described social activities during the preceding week: programs at community centers, religious activities, volunteer work, group activities at civic or community organizations, activities at senior-citizen centers, or regular sports activities with others. Each activity required a yes (1) or no (0) response and a single social participation score was calculated by summing responses to all six activities.

In this analysis, this study assessed the adult children’s age and financial transfers to support their parents (the grandparents). Age was measured by reported year of birth, and financial support to parents was measured by the amount of money the adult children had given to their parents in the past year, as reported by the parent (recipient of the support). Financial support was a continuous variable, and zero financial support was marked as 0.

Three variables representing family structure were included in the present study. Co-residence was categorized as either residing (1) or not residing (0) with adult children. Number of generations and number of people in the household were measures of the number of generations and people who lived together in the household.

### 2.3. Data Analysis

Predictors were used from the 2008 dataset (Wave 2) and the amounts of time spent caring for grandchildren in 2008, 2010, and 2012 (Waves 2–4) were used as the outcome variables. The first step in the analysis involved latent growth curves (LGCs) with initial levels and slopes as latent constructs. LGCs are suitable for studying individual differences in change and for understanding the process of change itself. These models utilize latent factors to estimate the fixed (group level) and random (individual level) components, and they also allow for separate trajectories over time on repeated measures. Each case in the sample can have a different intercept (initial level) and slope (change over time). The initial levels and slopes were determined by the amount of time spent caring for grandchildren. The initial levels were the amount of time reported at the first time point (Wave 2), and the slopes describe changes in that variable over time (as reported in Waves 3 and 4). This study then examined the changes in time spent caring for grandchildren by estimating the latent growth curves. The linear trajectories fit better than quadratic ones for the repeated measure (i.e., amount of time spent caring for grandchildren), and the resulting model reasonably fit the data: χ^2^(3) = 15.73, *p* < 0.001, comparative fit index = 0.95, and root mean square error of approximation = 0.04.

Then, using the growth parameters of the initial levels and slope constructs, this study utilized latent growth mixture modeling (GMM) to identify latent classes of longitudinal patterns of grandchild care. GMM is suitable for identifying latent classes of individuals, and maximum likelihood estimation is available to classify individuals based on their posterior class membership probability [39,40,41,42]. GMM further provides statistical methods for determining the optimal number of classes, or distinct clusters of longitudinal patterns in the data. This study used a few different model fit statistics to determine the optimal number of classes for this study. The Bayesian information criterion (BIC) is one of the best indicators to identify the optimal number of latent classes [43], and lower BIC values indicate a better model fit. Additionally, the Lo–Mendell–Rubin (LMR) test indicates whether model fit improved between k-class and (k-1)-class models [44].

Finally, to examine the relationships between the predictors and the longitudinal patterns of grandchild care, this study ran a multinomial logistic regression because the dependent variable was a three-level nominal measure derived from the GMM analysis as the optimal number of classes. This study also calculated the mean trajectories of grandchild care within each latent class. Longitudinal equation modeling using Mplus (Muthén & Muthén, Los Angeles, CA, USA) [45] was performed.

## 3. Results

### 3.1. Characteristics of Respondents

The analytical sample ranged from 50 to 70 years of age (*M* = 60.74, *SD* = 5.21) at Wave 2 and was comprised of 263 (79%) grandmothers and 70 (21%) grandfathers. Roughly 20% had completed high school, and the mean yearly household income in the sample was 26,160,000 KRW (roughly equivalent to $23,760 US dollars), which was lower than the mean yearly household income in South Korea (32,857,000 KRW) [46]. Approximately 26% of the sample was self-employed or paid for their work in 2008. In addition, grandparents reported receiving between 0 and 960,000 KRW from their adult children per month (*M* = 745,000 KRW, *SD* = 146,000), and this mean amount was roughly equivalent to $672 US dollars. Self-rated health status of grandparents tended to be good (*M* = 1.92, *SD* = 0.87), and social participation scores ranged from 0 to 4 (*M* = 1.17, *SD* = 0.84), indicating that many if not most respondents had participated in one or more of the queried activities within the preceding week.

Among the 333 grandparents, 181 (54%) cared for 1 grandchild, 134 (40%) cared for 2, and 18 (5%) cared for 3 or 4. The amount of time reportedly spent caring for grandchildren per week varied more within waves across grandparents than across waves within grandparents: Wave 2 data ranged from 1 to 98 h (*M* = 53.71, *SD* = 34.38), Wave 3 data ranged from 4 to 168 h (*M* = 57.45, *SD* = 45.45), and Wave 4 data ranged from 1 to 168 h (*M* = 47.42, *SD* = 38.73).

Respondents had a mean of 2.93 (*SD* = 1.14) adult children, all reported frequent interaction with their adult children, and nearly half (47%) lived with one of their adult children in 2008 (Wave 2). Their adult children ranged from 26 to 46 years of age (*M* = 36.0, *SD* = 4.37). Some respondents lived alone, and others reported sharing a household with up to eight people (*M* = 3.34, *SD* = 1.47) and four generations (*M* = 1.53, *SD* = 1.07). Although KLoSA did not collect information on whether grandchildren lived together with their grandparents, this study made the assumption that grandchildren lived with their grandparents when their parents lived with their grandparents. 

### 3.2. Growth Mixture Model with Three Trajectory Groups

The GMM produced a number of model fit statistics for determining the optimal number of classes (Table 1). The BIC decreased until reaching the three-class model and then increased for the four-class model, indicating that the three-class model fits the data the best. Further, the LMR was used to compare the three- and four-class models, and the two were not statistically different from one another. That finding, in conjunction with the smaller BIC for the three-class model, indicated that the three-class model was best (BIC = 4923.13, LMR = 35.55).

Figure 1 shows the model’s mean trajectories of the three classes. Class 1 (low-level decrease) showed a modest downward trend (slope change = −3.10, *p* = 0.003, as did Class 2 (high-level decrease; slope change = −4.21, *p* = 0.060), but Class 3 (low-to-high increase) showed a pronounced upward trend (slope change = 26.62, *p* = 0.000). Across the study sample, the classes ranged from 3.5% for the smallest class to 80% for the largest. More than three-quarters of the sample (*n* = 267, 80%) were assigned to the low-level decline group, which spent a relatively low amount of time caring for their grandchildren compared with the other groups (*M* = 40.9 h per week across the three waves). Moreover, the amount of time spent caring for grandchildren decreased over time from 44 h a week at Wave 2 to 37 h a week at Wave 4. The high-level decrease group was comprised of 55 grandparents (17% of the sample) whose time spent caring for their grandchildren also decreased slightly over time, but were distinct from the low-level decrease group in that they spent relatively high amounts of time caregiving for grandchildren (86 h a week at Wave 2 to 78 h per week at Wave 4). Finally, the low-to-high increase group consisted of a small number (*n* = 11, 3.5%) of grandparents who spent high amounts of time on grandchild care, and the amount of time increased over time (from 60 h a week at Wave 2 to over 110 h at Wave 4).

### 3.3. Predictors of Longitudinal Patterns of Grandchild Care

Table 2 shows the predictors related to these three longitudinal patterns of grandchild care. The low-level decline group was used as the reference group for comparison in the first two models. The grandparents who reported less social participation were more likely to be in the high-level decrease group or low-to-high increase group than in the low-level decrease group. Additionally, younger grandparents were more likely to be in the low-to-high increase group than in the low-level decrease group. It is notable that high school graduation and self-rated health status did not differentiate between the low-level decrease group and the other two groups. In terms of adult children factors, adult children’s age did not differentiate from the three classes, but grandparents who received financial support from their adult children were more likely to be in the high-level decrease group or low-to-high increase group than the low-level decrease group. Further, the number of people in the household was related to longitudinal patterns of grandchild care—grandparents who had fewer household members were more likely to be in the low-to-high increase group than the low-level decrease group. In summary, grandparents’ age, gender, and social participation, financial support from adult children, and household size differed among the three classes.

To complete the comparisons between groups, this study also compared the high-level decline group with the low-to-high increase group (reference group). In this model, only adult children’s age distinguished the two groups: grandparents with older adult children were more likely to be in the high-level decrease group.

## 4. Discussion

Using general growth mixture modeling, this study investigated longitudinal patterns of grandchild care to examine the factors that predicted the longitudinal pattern of grandchild care in South Korea. This study found three classes of grandchild care based on group means: low levels of caregiving and decreasing (Class 1), high levels of caregiving and decreasing (Class 2), and low levels of caregiving followed by a rapid increase (Class 3). These distinct subgroups of older adults who follow unique trajectories of grandchild care may be distinct in some important but as yet unknown ways. This study found differences in the predictors of group membership, but those differences were usually small in magnitude.

Using group-based modeling, the majority (80%) of older adults who cared for their grandchildren provided low levels of caregiving that declined over time. Other trajectories (i.e., high levels of caregiving that declined over time, and low levels of caregiving that increased precipitously over time) were far less common, but nonetheless comprised nearly 20% of our sample. It is noteworthy that even the grandparents in the low-involvement group (i.e., low-level decrease group) were still providing the equivalent of full-time care based on the mean number of hours of care provided per week (40.9 h). Thus, although most grandparents in the larger sample were not caregiving for grandchildren, those who were caregiving for grandchildren were investing a great deal of time into doing so. 

Given our inclusion criteria, the disproportionate number of grandmothers relative to grandfathers in this study supports previous findings indicating that grandmothers are more likely than grandfathers to provide grandchild care [5,22,27]. According to related research, grandmothers, as a stereotype of gender roles in South Korean society, maintained their identity as a primary caregiver who raised their children in the past, and they perceived that they should nurture their grandchildren rather than grandfathers [47]. Parenting is still considered to be a part of women regardless of age, and it is an important context for supporting grandchildren [47]. Thus, society should constantly monitor grandmothers who care for their grandchildren.

In this study, grandparents who provided more grandchild care also tended to receive more financial support from their adult children. This relationship might be explained in the South Korean context using the social exchange theory. That is, this finding supports the intergenerational exchange theory that grandparents transfer time to their adult children in exchange for financial support [1,48]. Previous research has found that grandparents who care for grandchildren are less financially prepared for self-sufficiency later in life [16,17], perhaps because they invest in their grandchildren rather than securing resources for themselves. Therefore, one explanation is that older parents provide care for grandchildren in order to receive financial support and caregiving from their adult children when they can no longer take care of themselves. Alternatively, it may be that grandparents caring for grandchildren and adult children caring for older parents are simply behaviors that co-occur in close families. However, this study was unable to examine the latter because family closeness was not measured in the KLoSA data.

## 5. Implications

The results of this study can inform policymakers and practitioners in the design of later-life preparation programs for older people. In the past, South Korean cultural norms dictated that adult children were socially obligated to care for their older parents, but the veracity of that expectation has waned over time [49]. Rather, many adult children in South Korea now believe that older parents are responsible for taking care of themselves in later life, meaning that they are expected to be prepared for later life in advance [17,48]. Considering that caring for older adults typically falls upon the government when self-sufficiency is not possible and informal support systems fail, it would behoove the government to ensure that mechanisms are in place that will provide older adults a means to maintain their self-sufficiency, especially in the case of grandparents caring for their grandchildren to support working parents when other childcare options are unsuitable.

The relationship between social participation and grandchild care is likely bidirectional, but our findings suggest that grandparents may be providing grandchild care in lieu of participation in social activities, as has been found in other studies [1]. This is not a surprising finding because there are obviously fewer opportunities for older people to participate in social activities when routinely engaged in the time-consuming role of caretaker for a grandchild. Thus, caring extensively for grandchildren may work against grandparents’ ability to maintain a socially active lifestyle, and this may be problematic because social participation is associated with healthy aging [50]. Communities should work to counter this by developing programs that facilitate participation in social activities among grandparents who care for their grandchildren. For example, creating opportunities for grandparents within a community who are caring for grandchildren to congregate and communicate with one another could be both appealing and useful.

There is also a shortage of qualified childcare facilities available in South Korea. South Korean working mothers who need daycare services for their children often complain that it is hard to find reliable care facilities [51]. Consequently, in lieu of formal childcare services, many grandparents have started providing childcare for their grandchildren. The government can play a vital role in ensuring that there are a sufficient number of reliable childcare facilities available to meet working parents’ needs. For example, a simple but impactful change would be keeping public childcare facilities open later to accommodate the schedules of parents who are employed.

Family life educators might also play a role in helping to alleviate the burden of grandparent caregiving by developing programs designed to educate grandparents and their adult children about the potential burdens and challenges of such arrangements, as well as to offer strategies for success. The development of respite care programs for grandparents who are providing care to grandchildren would also help to preserve the health and well-being of individuals as well as families. Similarly, counseling programs would allow grandparents to share emotional distress, and in doing so may help to both minimize the impact of stress and maintain health [52].

## 6. Limitations and Future Directions

This study has several limitations that must be taken into account when considering the findings. In particular, several pieces of missing information in this secondary dataset limited our ability to examine all relevant pieces of information. For example, this study does not know the employment status or longitudinal patterns of the grandchildren’s mother, and evidence suggests that grandchild care in South Korea may vary according to the mother’s employment status [1]. Also missing were factors related to formal childcare, such as whether the grandchildren ever attended community daycare centers or received childcare services at home. Additionally, this study was unable to determine who the primary caregiver of grandchildren was when both grandmother and grandfather indicated that they provided grandchild care. This study was also left to assume that grandchildren lived with their grandparents if the grandparents reported living with their adult children. In addition, wide variation in the amount of time reportedly spent caring for grandchildren may indicate that the question was answered differently by different people (e.g., if living with a grandchild, some grandparents may have included sleeping time and time where parents are also present, whereas others may have limited their response to time spent alone with or directly supervising the grandchild). This study limited grandparents who reported providing grandchild care for at least one hour per week across each of the three waves. Finally, it was notable that only a small proportion of the sample (low-to-high increase group, 3.5%) spent more hours per week grandparenting as they grew older. Future research needs to identify longitudinal patterns of grandchild care through various data and research methods. In addition, the use of data from 10 years ago in this study is a limitation. In the future, more studies are requested to proceed with related studies. 

These limitations might have influenced the study findings, so as is always the case, our findings should be interpreted with due caution until replicated with other samples. In addition to replication, future studies should advance this body of literature by more carefully examining grandchild care in the context of adult children’s employment status and schedule, as well as access to community childcare services. The complex interplay of grandparents’ motivations for providing grandchild care in the larger context of intergenerational resource transfers should also be examined in greater detail. Parsing out the role of socioeconomic status in caregiving trajectories and motivations may also be fruitful for better understanding families’ needs. 

## 7. Conclusions

Given the lack of longitudinal research in South Korea on grandparents who provide grandchild care, this study makes an important contribution to the current knowledge base. Our study provided a first step toward defining distinct subgroups among grandparents who care for their grandchildren, and in doing so began a process of developing an empirical foundation that may have implications for improving the experiences and outcomes of grandchild care, as they have been increasingly doing over the past few decades in conjunction with women’s movement into the workforce and inadequate formal childcare options. In South Korea and many other developed and emerging economies, where modern norms are for mothers to be employed, identifying the stressors, needs, and outcomes associated with this shift will be an important next step toward building and maintaining productive workforces, healthy societies, and functional families. In South Korea, the increasing involvement of grandparents in caring for their grandchildren presents both problems and possibilities that are worthy of further exploration.

## Figures and Tables

**Figure 1 ijerph-19-01136-f001:**
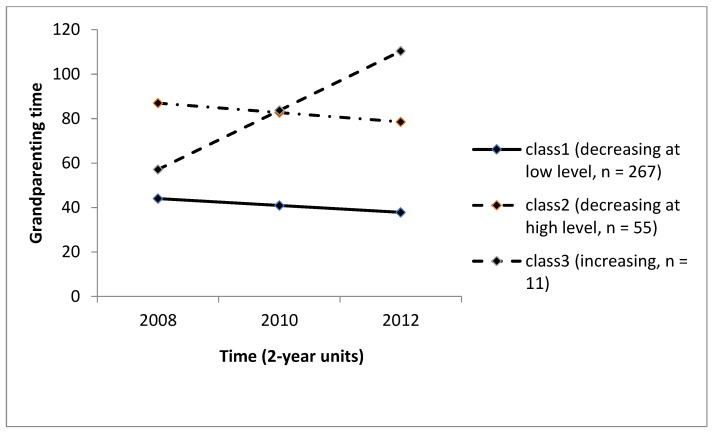
Three longitudinal patterns of grandchild care.

**Table 1 ijerph-19-01136-t001:** Latent class analysis model fit statistics.

Number of classes	AIC	BIC	Entropy	LMRT
Class 1	4975.83	4998.73	N.A.	N.A.
Class 2	4895.35	4941.15	0.85	74.25 *
Class 3	4865.88	4923.13	0.71	33.55 †
Class 4	4871.88	4940.59	0.57	0.04

Note: ACI = Akaike Information Criteria, BIC = Bayesian Information Criteria, LMRT = Lo–Mendell–Rubin Likelihood Ratio Test; † *p* < 0.10; * *p* < 0.05.

**Table 2 ijerph-19-01136-t002:** Multinomial logistic regression: predictors and longitudinal patterns of grandchild care.

	Class 2 (High-Level Decrease) ^Class 1^					Class 3 (Low-to-High Increase) ^Class 1^					Class 2 (High-Level Decrease) ^Class 3^				
Predictor	*B*	*SE*	*p*	*OR*	95% CI	*B*	*SE*	*p*	*OR*	95% CI	*B*	*SE*	*p*	*OR*	95% CI
Grandparent factors															
Age	0.01	0.09	0.930	1.01	[0.85, 1.20]	−0.11	0.42	0.011	0.90	[0.83, 0.98]	−0.12	0.09	0.196	0.89	[0.75, 1.06]
Male (female)	−1.84	1.25	0.142	0.16	[0.01, 1.85]	−1.10	0.53	0.037	0.33	[0.12, 0.94]	0.74	1.24	0.549	2.10	[0.19, 23.64]
Graduated from high school (did not)	−0.77	0.85	0.362	0.46	[0.09, 2.43]	−0.20	0.44	0.651	0.82	[0.35, 1.93]	0.58	0.84	0.494	1.78	[0.34, 9.30]
Self-rated health	0.05	0.41	0.905	1.05	[0.47, 2.33]	0.16	0.22	0.468	1.17	[0.76, 1.80]	0.11	0.41	0.785	1.12	[0.50, 2.48]
Social participation	−1.44	0.69	0.038	0.24	[0.06, 0.92]	−0.65	0.31	0.037	0.52	[0.29, 0.96]	0.80	0.69	0.252	2.22	[0.57, 8.65]
Adult children factors															
Age	−0.16	0.10	0.107	0.85	[0.70, 1.04]	0.06	0.05	0.211	1.06	[0.97, 1.17]	0.22	0.10	0.032	1.25	[1.02, 1.53]
Financially supported parents (did not)	0.01	0.003	0.000	1.01	[1.01, 1.02]	0.01	0.002	0.000	1.01	[1.01, 1.02]	0.00	0.002	0.350	1.00	[0.99, 1.01]
Family structure factors															
Co-resided (did not)	0.06	0.35	0.857	1.07	[0.54, 2.12]	0.09	0.19	0.647	1.09	[0.75, 1.58]	0.02	0.35	0.948	1.02	[0.51, 2.04]
Generations in household	0.04	0.34	0.917	1.04	[0.53, 2.02]	−0.18	0.23	0.425	0.83	[0.53, 1.31]	−0.18	0.36	0.549	0.80	[0.39, 1.64]
People in household	−0.15	0.54	0.789	0.86	[0.30, 2.49]	−0.78	0.34	0.021	0.46	[0.24, 0.89]	−0.63	0.59	0.286	0.53	[0.17, 1.70]

Note. Reference category in parentheses. Class 1 = low-level decrease. CI = confidence interval for odds ratio (*OR*).

## Data Availability

Information on how to obtain the KLoSA data files is available on the KLoSA website (https://survey.keis.or.kr/klosa/klosa01.jsp) accessed on 30 January 2017.

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
