# Peer review of "Longitudinal Patterns of Grandchild Care in South Korea"

_ijerph, 2022, doi:10.3390/ijerph19031136_

Round 1

Reviewer 1 Report

This study is highly significant in the field of early childhood, family studies, the aging population, whilst tapping into multiple societal and cultural factors including child-care arrangements, social and emotional impact for grandparents, financial aspects, and more. I feel as though this study has opportunities to investigate these issues further in future publications, as it raised numerous and notable aspects of the South Korean society. Although the focal point of this study was not the children per se, I would also be interested to read the impacts of this study on the children's development (the socio-cultural phenomenon or impact on their learning, education, growth, etc) when cared for by grandparents. Furthermore, within the South Korean culture, it may even be 'expected' for grandparents to take-on the care-giving role for working parents. The emotional pressure for grandparents to care for their grandchildren, is an interesting perspective that could be discussed further. Lastly, as the author has noted in the paper, the quality of the early childhood centres may limit choices for parents which I thought was a brilliant point. This was a pleasant read and I look forward to reading more on this important research. 

Reviewer 2 Report

Dear Author(s),

I found your topic and data are very interesting, but the manuscript contains serious problems that should be fixed before considering as a publishable piece of work in IJERPH.

For instance, 

It is hard to follow the relationship between social exchange theory (SET) and your research questions. SET is one of the best well-developed, empirically-tested, formal theories in the literature. If you are referring formal theories, such as SET, you are expected to utilize its most advanced versions to structure your research design, including developing your derivations and hypotheses. Then, the paper would add a new test for the theory. Or you may develop the theory through integrating other theories or research. Generally, we expect a researcher to tell us a new relationship(s) among factors, underlying mechanisms, etc. to better understand the phenomenon.

So the paper is not providing us empirical data demonstrating/testing assumptions from social exchange theory.

As expected, the literature review is very limited in terms of theory.

If the paper's argument not built on an appropriate base of theory, concepts, or model, then theory building is not successful and thus empirical operationalization and testing procedure do not make very much sense either.

Since the theoretical claims were not clear, it is hard to figure out what the demonstrated data testing is.

Overall, if you may fixed problems in your data such as the distribution gap of your sample among subgroups, clarify the purpose of these subgroups and connections with the predictors, etc. you may just analyze the data, explore the different relationships and report your findings. 

Good luck

Reviewer 3 Report

This manuscript reports a research that identifies and analyses the longitudinal patterns of time spent by grandparents caring for their grandchildren in a South Korean national sample. The study was conducted in the scope of a larger national survey which is a strength but also implies some limitations that are acknowledged by authors.

I find the study interesting and well supported conceptually. The method is adequate considering the objectives of the study, the results are clearly presented and the discussion is relevant and well structured. However, authors should address some imprecisions and inconsistencies before the manuscript is considered for publication. I pinpointed the aspects that deserve further attention and present some suggestions, following the order of the manuscript.

1) From my point of view, the title does not capture the essence of the study and is not completely accurate. Since the focus of the manuscript is the longitudinal patterns of time spent by grandparents caring for their grandchildren, a simpler alternative could be for instance “Longitudinal patterns of grandchild care in South Korea”.

2) In the abstract and throughout the manuscript, authors refer to the longitudinal patterns in different ways: “longitudinal classification of time changes”, “latent classes of longitudinal patterns in time spent …”, “longitudinal patterns of care”, “latent trajectories”, “classes of time spent caring …”, “categories”. Using a more consistent terminology would help readers improve their understanding of the study.

2) The titles of the classes are not presented consistently. For instance, in the abstract, classes are named as “decreasing at a low-level decline”, “decreasing at a high-level decline”, and “increasing from low-to-high increase”; while on page 8 they are named as “low-level decrease”, “high-level decrease”; “increase from low to high” and on Table 2 this last one is named as “low to high increase”. From my point of view, the simpler, the better.

3) Whenever possible, authors should use precise language /terminology. For example in the abstract, “of a number of factors” (how many? which factors?), “Substantial” (what does it mean?). The same vague terms are used in the first paragraph of the discussion section (“well-established”, “diverse”). In the same paragraph, please note that the three classes of grandparents are different not in the amount of time spent caring for grandchildren but in the LONGITUDINAL patterns of time spent in grandchild care, and this is a quantitative difference (not qualitative).

4) The breaks in the following sentence disrupts the reader’s comprehension of the message: “Social participation has been found to be associated with a higher likelihood of providing grandchild care (than not) and with providing more (than less) grandchild care”

5) In spite of mentioning South Korea in the title as the country where the study was conducted, all over the manuscript authors refer to Korea only. I suggest the use of the official name of the country.

6) I understand that the increase of grandchild care, as well as other forms of formal and informal childcare, is related to the increase of women’s participation in the job market. However, it is more difficult for me to understand extra-familial care as a response to working mothers’ needs. Unless this is a cultural issue that I am not aware of, I would said that extra-familial care is a response to dual-earner families’ needs, as both parents have a professional career and no conditions to take care of their children on a full-time basis.

7) Authors discuss some relevant limitations of the study. However, three additional limitations deserve attention:

(1) Study participants correspond to grandparents that took care of grandchildren during the four years of the project (i.e., across each of the three waves). Consequently, participants that provided grandchild care in less than the three waves were excluded from the sample. The rational for this decision is not presented.

(2) Data were collected in 2008, 2010 and 2012, more than a decade ago. Even if we do not consider the present pandemic scenario, how do the results reflect the present situation?

(3) The three classes that were extracted have a quite uneven number (267, 55, 11). How was this considered in data analysis?

I hope the authors find my comments and suggestions useful.
